# Does cotrimoxazole prophylaxis in HIV patients increase the drug resistance of pneumococci? A comparative cross-sectional study in southern Ethiopia

Mohammed Seid[1]*, Getnet Beyene[2], Yared Alemu[2], Bereket Workalemahu●[3], Mulugeta Delbo[3], Dagimawie Taddesse[1], Gelila Biresaw[1], Aseer Manilal●[1]*

1 Department of Medical Laboratory Science, College of Medicine and Health Sciences, Arba Minch University, Arba Minch, Ethiopia, 2 Department of Medical Laboratory Sciences and Pathology, Jimma University, Jimma, Ethiopia, 3 Department of Medical Laboratory Science, Arba Minch College of Health Sciences, Arba Minch, Ethiopia

* mohamedseid2005@gmail.com (MS); aseermanilal@gmail.com (AM)

## Abstract

### Background

Infections caused by antibiotic-resistant bacteria results in high rates of morbidity and mortality. Although the prolonged cotrimoxazole (CTX) prophylaxis is arguably associated with the risk of increasing drug resistance in the common pathogens, information regarding its impact on *Streptococci pneumoniae / pneumococcus* is very limited.

### Objective

This study was conducted to investigate the effect of cotrimoxazole prophylaxis on nasopharyngeal colonization rate and antimicrobial resistance using *Streptococci pneumoniae* (pneumococcus) as an indicator organism among HIV patients in Arba Minch, Ethiopia.

### Materials and methods

A comparative cross-sectional study was designed and conducted among HIV patients attending the Anti-Retroviral Treatment (ART) clinic of Arba Minch General Hospital (AMGH) from April 01 to August 31, 2018. A total of 252 participants were systematically selected and clustered into two study groups based on their CTX prophylaxis status, one taking CTX prophylaxis, and the second one, the control group (without prophylaxis). A structured questionnaire was used to collect socio-demographic and clinical data from patients. A nasopharyngeal swab was collected and cultured for pneumococcal isolation and identification in accordance with standard microbiological techniques. An antibiotics sensitivity test was performed according to the CLSI guidelines. Data were analyzed using the Statistical package for social science (SPSS) version 20. The primary outcome was determined using logistic regression analysis.

**Data Availability Statement:** All relevant data are within the manuscript and its Supporting Information files.

**Funding:** The author(s) received no specific funding for this work.

**Competing interests:** The authors have declared that no competing interests exist.

## Results

Of the 252 enrolled HIV patients (mean age (37.38± 9.03 years), 144 (57.14%) were males. The overall, nasopharyngeal colonization rate of *S. pneumoniae* was 13.5% (95% CI: 8.4–15.6). Asymptomatic pneumococcal carriage rates among patients on CTX prophylaxis and the control group were 16.3%, and 10.3% respectively (p-value = 0.03). Regarding the risk factors analyzed, CTX prophylaxis (AOR: 2.2; 95% CI: 1.05–4.9) and gender (AOR: 2.5; 95% CI: 1.09–5.93) were significantly associated with pneumococcal colonization, showing a male preponderance. Cotrimoxazole-resistant pneumococci were 85.7% vs. 47.4% in the prophylaxis group and the control group respectively and it was statistically significant (AOR: 6.7; 95% CI: 1.3–36). Percentages of multi-drug resistant isolates in these two groups were 38.09 and 15.38 respectively (p-value = 0.04). Among the CTX resistant pneumococci isolates, 85% were also found to be co-resistant towards penicillin and was statistically significant.

## Conclusion

The percentage prevalence of nasopharyngeal pneumococci colonization was higher in patients taking CTX prophylaxis. It was noted that CTX prophylaxis eventually results in the selection of cotrimoxazole resistance and multi-drug resistance in pneumococci. There is evidence of existing cross-resistance between cotrimoxazole and penicillin antibiotics. Therefore, CTX prophylaxis must be administered judiciously. Surveillance for antimicrobial susceptibility is warranted where the prophylaxis is common.

## Introduction

Cotrimoxazole prophylaxis is effective in reducing the morbidity and mortality in children and adults infected with Human Immunodeficiency Virus (HIV), however, its impact on the risk of increasing antimicrobial resistance remains a debatable public concern globally [1, 2]. Cotrimoxazole (CTX) is a broad-spectrum antibiotic co-formulation of trimethoprim and sulfamethoxazole (TMP-SMX), which inhibits the synthesis of bacterial tetrahydrofolic acid. Currently, it is used for the management of common bacterial infections and preventive prophylaxis against opportunistic infections among HIV patients [3]. Although World Health Organization (WHO) and the Joint United Nations Program on HIV/AIDS (UNAIDS) issued recommendations regarding the cotrimoxazole prophylaxis in sub-Saharan Africa based on the evidence from trials done in Cote d'Ivoire since 2000, implementation of the policy was not that effective, especially in many resource-limited countries [2, 4–6]. The ineffective implementation was partly due to the uncertainty in the adverse microbiological consequences following the widespread use of CTX, inadequacy of data, considerable gaps in the coverage, and the magnitude of impact, at the individual as well as massive levels [7].

Antimicrobial resistance is a growing global problem that causes enhanced death rates as well as increased suffering. Overall, two main factors drive antimicrobial resistance: the volume of antimicrobials used and the spread of resistant micro-organisms along with the genes encoding the resistance [8, 9]. The widespread CTX therapy upsets the patient's micro-flora and results in the selection of resistant strains among commensals and pathogenic organisms at every ecological niche: patient, community, region, or country [10, 11]. Selective pressure, which is defined as the influence exerted by the use of antibiotics aiding the stepwise selection

of resistance by killing susceptible bacteria [9, 10]. To date, there are two opposing views on the inescapable adverse microbiological consequences of widespread CTX prophylaxis. One view is that the selective pressure results in bacterial resistance for the drug itself among patients on treatment, which should be avoided from occurring in the future course [10, 11]. An alternative view is that resistance shown by pathogens against CTX prophylaxis is uncertain or evidence is nominal only, and hence there is no need for any action [12, 13]. Evidence from previous studies also showed variable outcomes. It is envisaged that CTX prophylaxis increase the risk of resistance by common pathogens [7, 8]. Recent studies also highlighted that CTX prophylaxis increase the risk of resistance shown by common clinical isolates such as *S. pneumoniae* [14] *Escherichia coli* [15], *Staphylococcus aureus*, *Haemophilus influenzae*, Enteropathogens (*Shigella* spp., *Salmonella* spp. and *Neisseria* spp. [6, 16, 17]. However, in contrast to this, a couple of studies reported that CTX prophylaxis does not affect the emergence of resistance and colonization rates, helping to protect from common bacterial pathogens [7, 15]. In particular, CTX prophylaxis may induce bacterial cross-resistance to other classes of antibiotics and associated risk of multidrug resistance (MDR). Biologically this is plausible because of the mechanism of linked resistance and horizontal transfer of resistant genes among bacterial species and genus [7, 18–20]. Hence the increasing drug resistance extends beyond CTX resistance to MDR [6–8]. There is some inconsistency among studies, partly reflecting a significant heterogeneity in terms of design, the number of participants, study population, and the duration of follow-up. Thus drawing conclusions based on earlier studies is problematic [7, 15, 21]. Antibiotic resistance patterns may vary locally and regionally, so surveillance data need to be collected from selected sentinel sources [2, 6, 8]. In this context of the increasing threat of antibiotic resistance, we have chosen *S. pneumoniae* as a sentinel indicator organism to assess the effect of CTX prophylaxis due to the following reasons. First of all, nasopharyngeal colonization can be considered as a surrogate for invasive diseases [22, 23]. *S. pneumoniae* is one of the major pathogens responsible for high rates of diseases, death, and permanent injury in children and adults globally [24–26]. Secondly, nasopharyngeal pneumococcal colonization is an important reservoir and it acts as a marker for resistance among co-colonizing microflora in the upper respiratory tract of the host (e.g., *H. influenzae* and *N. meningitides*) [27–29]. Thirdly, a high prevalence of resistant pneumococci is identified by WHO as one of the most pressing issues [30–33]. Even conflicting reports on nasopharyngeal pneumococcal colonization are pouring in from different countries across the globe and will be elaborated later in this paper. Lastly, so far there is no surveillance system existing in Africa, to manage and limit the spread of this menace [33, 34]. Hence this comparative cross-sectional study was conducted in Ethiopia, where an estimated 0.72 million people were living with HIV [13] and CTX prophylaxis is in practice for more than 15 years. The main objective of this study was to test the hypothesis that CTX prophylaxis increases the colonization rate and antimicrobial resistance among HIV patients. The site selected for this purpose is Arba Minch General Hospital, southern Ethiopia. This study makes two important contributions. First of all, it provides new insights into CTX prophylaxis and drug resistance. Secondly, these findings could be used as baseline data for further studies, connected to CTX prophylaxis in the treatment of acute respiratory tract infections.

## Materials and methods

### Design of study

A comparative cross-sectional study was carried out from April 01 to August 31, 2018, at AMGH, southern Ethiopia. The hospital is situated in Gammo Zone, Southern Nations, Nationalities' and Peoples' Region and located at 505 km south of Addis Ababa, the capital city

of Ethiopia. AMGH was established in 1969 and now it serves a population of more than two million. In Ethiopia, the free ART program was launched in 2005 and AMGH became part of this scheme since its inception.

## Study participants

All registered HIV patients ≥15 years of age, who were attending follow-up visits at the ART clinic of AMGH, between April 01 and August 31, 2018, were eligible for enrollment in the present study. They were systematically selected and enrolled either in the study arms of the CTX prophylaxis group or in the control group by checking their prophylaxis status. Patients taking CTX prophylaxis constituted the first group and those who were not taking it represent the control group. Members of the former group were consuming daily CTX (800 mg sulfa-methoxazole, 160 mg trimethoprim) as a double-strength tablet, i.e. single tablet of 960 mg /day or two single strength 480 mg tablets/day for a minimum of two weeks duration, against opportunistic infections.

All enrolled patients met the following inclusion criteria: aged above 15 years, with complete medical records, not severely ill, and willing to participate. However, inpatients, clients with incomplete and unclear or lost medical records, those who are on CTX prophylaxis for only less than two weeks or with a history of discontinuance (poor adherence), were excluded from both study groups. Subjects who had received antibiotics known to be active against *S. pneumoniae* within the previous two weeks of the commencement of the study were also excluded. There could be some under-reporting of the usage of antibiotic, but we would expect this to be distributed approximately equally among subjects taking CTX prophylaxis and the control group, as a result of random selection.

## Sample size determination and sampling technique

We enrolled 252 HIV patients, depending on the prevalence of nasopharyngeal colonization and CTX resistance to pneumococci that have been observed from a previous study [16]. This helped to measure a 20% difference in the prevalence of nasopharyngeal colonization rate and CTX resistance among HIV patients taking CTX prophylaxis and the control group, with 80% power at 95% confidence level ($\alpha = 0.05$). With the above assumptions, the required sample size for the study was determined employing two-population proportion formula using the statistical program, Epi-Info, CDC, Atlanta, GA computer software statistical package. The overall sample size of 252 HIV patients was then enrolled in the study arms in a 1:1 ratio (ie., n1 = 126 and n2 = 126). The study unit was organized using a systematic random sampling method, from each study population belonging to the respective group, after the complete listing (sampling frame) of active follow up of the registry, which had been prepared in chronological order. The sampling intervals were fixed based on the aforementioned study population and the calculated sample size. Fortunately, almost equal numbers of patients were finally found to be included in both the CTX prophylaxis group and the control group.

## Data collection and laboratory processing

After obtaining the informed consents, socio-demographic and behavioral factors including age, sex, religion, education, family size, smoking status, and contacts in sleeping rooms (with children attending daycare centers) were collected by a predesigned and pre-tested questionnaire, through face-to-face interviews [16, 35]. The questionnaire was translated into Amharic, the local language, and back-translated to English by independent translators to ensure the exact meaning of each word. Clinical and laboratory data that include $CD_4$ T cell counts, history of upper and lower respiratory tract infections, details of CTX prophylaxis (dosage &

duration), latest (within the last two weeks) usage of antibiotics, clinical stages of the disease, history of hospitalization and the highly active antiretroviral therapy (HAART), were obtained from the patients' medical records and then entered in the collection checklist [16].

## Isolation and identification of *S. pneumoniae*

The nasopharyngeal specimen of each patient was collected by trained study personnel using sterile calcium alginate swabs (Fisherbrand, Fisher Scientific, Pittsburg, PA). This was done by tipping the patient's head slightly backward and passing the swab directly upward, parallel to the floor of the nasopharynx. The swab was passed till it reached the posterior pharynx. Once in place, the swab was rotated through 180 degrees and left in place for two to ten seconds to saturate the tip. It was then immediately placed in 1.0 ml of skim milk tryptone-glucose-glycerin transport medium and stored at -80˚C until inoculation onto a blood agar plate. The swab was inoculated onto tryptone soya blood agar (Oxoid, Basingstoke, Hampshire, England) supplemented with 5μg gentamicin and incubated at 37˚C under 5% $CO_2$ atmosphere overnight (16–20 hours). The plate with no growth after 24 hours of incubation was further incubated for additional 16–20 hours. Isolates of *Streptococcus pneumoniae* were identified according to WHO guidelines which include hemolytic property, Gram's staining, catalase, optochin sensitivity, and bile solubility tests [35, 36].

## Antimicrobial susceptibility testing

The antibiotic susceptibility profile was determined by the Kirby-Bauer disk diffusion technique according to the criteria set by the Clinical Laboratory Standard Institute (CLSI), [36]. Seven commercially available antibiotic discs (Oxoid, Basingstoke, Hampshire, UK) were used, viz., oxacillin (l μg), chloramphenicol (30 μg), erythromycin (15 μg), tetracycline (30 μg), cotrimoxazole (25 μg), levofloxacin (5 μg), and vancomycin (30 μg). Inoculums were prepared by picking parts of similar test organisms with a sterile wire loop and suspending them in sterile normal saline. The density of suspension to be inoculated was determined by comparison with an opacity standard on McFarland 0.5 barium sulfate solution. The test organisms were uniformly seeded over the Mueller Hinton agar supplemented with 5% sheep blood plates. Six antibiotic discs were dispensed on the seeded lawn at 60˚C sufficiently apart from each other and incubated at 37˚C in 5% $CO_2$ for 24 hours. Minimum inhibitory concentrations (MICs) for penicillin were determined for all oxacillin non-susceptible isolates by the Epsilometer test (Biomerieux, Durham, NC). MIC results were interpreted according to the CLSI guidelines [36]; breakpoint and intermediate levels were considered as the resistance.

## Ethics approval and consent to participate

Permission to carry out this study was obtained from Jimma University School of Medicine, Research and Ethical Review Board (IRB/2531/2018). Written consent/assent was sought and obtained from each study subject or guardian, consistent with the Helsinki Declaration relating to the conduct of research on human subjects.

## Quality control

Prior to the data collection, training was given to data collectors. A pretest was conducted in Chencha Hospital, Arba Minch, Ethiopia on 5% of study participants to assure that the data collection format is feasible in closely related settings. A standard operating procedure was followed. Quality control of susceptibility patterns was assured on a daily basis throughout the

study, using *S. pneumoniae* (ATCC 49619) and *S. aureus* (ATCC 25923). Susceptibility results were interpreted according to the guidelines of CLSI [36].

## Statistical analysis

Data were entered into Epi-data version 3.1 and exported and analyzed by using SPSS software version 20 (IBM Statistics, Armonk, NY, United States). Proportions were calculated for categorical variables and summaries were presented in terms of counts and percentages. Prevalence of pneumococcal colonization was compared between the groups of patients taking CTX prophylaxis and the control. Bivariable analyses were performed for each variable independently. Logistic regression analysis was done to determine the independent predictors of pneumococcal colonization rate. Then, to control simultaneously the possible confounding effects of different variables that may affect the results, the risk of having cotrimoxazole prophylaxis on colonization was estimated by multivariate analysis. Variables considered were sex, age, $CD_4$ T cell counts, clinical stages of the disease defined by WHO, co-morbidity, hospitalization, crowding, and antiretroviral therapy. Additionally, to explore whether the CTX prophylaxis contributes to the resistance profile of pneumococci or not, the prevalence of resistance to CTX, penicillin, and MDR were compared among isolates from patients taking CTX prophylaxis and the control group. In order to explore the influence of CTX prophylaxis on the cross-resistance profile of pneumococci, the prevalence of resistance to one or more antibiotics other than CTX was compared among CTX–resistant and susceptible isolates. P values < 0.25 in the bivariable analysis were further included in the multivariable analysis. The degree of association between a dependent and independent variable was assessed using an odds ratio, with a 95% confidence interval. All tests were two-tailed, and a p-value of < 0.05 was fixed as a cutoff point to determine the presence of a statistically significant association.

## Results

### Baseline characteristics

A total of 252 subjects were enrolled and then bifurcated, creating a pair of study arms: 126 HIV patients who have been on CTX prophylaxis, and 126 HIV patients who were not on CTX prophylaxis (the control group); age of the study subjects ranged from 16 to 67, the mean being 37.38± 9. The age group, 30–44, with a total of 163(64.68%) constituted the majority. HIV-infected adults, among patients in the CTX prophylaxis group were slightly younger (37 ± 9) than those in the control group (38.1± 10), according to the mean age. The proportion of male participants were higher in the former group, ie., 63.5% against 50.8% in the control group. The majority of participants (91.27%) are from urban areas. The ability of participants to read and write was limited; 20.63% was illiterate, 29.76% had only primary education, 23.4% had attended secondary schools, whereas 21% went to high schools and 4.37% only had received the tertiary education (Table 1).

Fourteen (5.5%) of the study participants had a history of hospitalization within the previous three months to the commencement of the study period. The majority of HIV patients were on antiretroviral therapy [241 (95.63%)]. Most of those who were not on CTX prophylaxis was on WHO clinical stage I, whereas the majority of those who were on CTX prophylaxis were on stage II, stage III, or even some of them are on stage IV. Overall, $CD_4$ T cell counts of HIV patients ranged from 31–1450 cell/ $mm^3$, the mean $CD_4$ T cell counts being 591± 13 cell/ $mm^3$.

### Nasopharyngeal colonization rate and associated factors

Out of the total 252 nasopharyngeal samples collected, 34 were culture-positive for *S. pneumoniae* giving an overall pneumococcal colonization rate of 13.5% (95% CI: 8.4 to 15.6).

**Table 1. Socio-demographic characteristics of HIV patients undergoing CTX prophylaxis compared with the control group from April 01 to August 31, 2018, Arba Minch, Ethiopia (n = 252).**

| Variables | | Total No. (%) | No. of participants (%) | | P-value |
|---|---|---|---|---|---|
| | | | CTX prophylaxis (n = 126) | Control group (n = 126) | |
| **Age in years** | 16–29 | 35(13.8) | 18(14.28) | 17(13.49) | 0.65 |
| | 30–44 | 163(64.6) | 83(65.87) | 80(63.49) | |
| | 45–59 | 52(20.6) | 24(18.75) | 28(22.22) | |
| | ≥60 | 1 | 1 | 0 | |
| **Sex** | Male | 144(57.1) | 80(63.49) | 64(50.8) | 0.30 |
| | Female | 108(42.8) | 46(6.5) | 62(49.2) | |
| **Residence** | Urban | 230(91.2) | 113(89.7) | 117(92.9) | 0.01 |
| | Rural | 22(8.7) | 13(10.3) | 9(7.1) | |
| **Education** | Illiterate | 52(20.6) | 21(16.67) | 31(24.2) | 0.09 |
| | Primary | 75(29.7) | 38(30.13) | 37(29.4) | |
| | Secondary | 59(23.4) | 36(28.57) | 23(18.3) | |
| | High school | 58(21.8) | 26(20.63) | 29(23.0) | |
| | College and above | 11(4.3) | 5(3.97) | 6(4.8) | |

Pneumococcal colonization was the highest 20(12.3%) in the age group 30–44 and much lower in participants whose age was > 60. Out of the total male participants, 26(18%) were carriers. Of the respondents from the urban area, 28(12.2%) were colonized with *S. pneumoniae*. In the case of participants, who were active cigarette smokers, only 3(17.6%) were carriers, whereas 6(12.2%) members who had frequent contacts with children aged <5 years in the house were found to be colonized with pneumococci. Surprisingly, a higher nasopharyngeal colonization rate of 22(18.8%) was reported among individuals who had a small family size (Table 2).

The pneumococcal colonization rate was 32(13.3%), among patients who were on ART. Pneumococcal colonization was slightly higher in patients who were on prophylaxis compared to the control group (16.6 vs. 10.3%). Bivariate analysis showed that age, family size, history of hospitalization, respiratory tract infections, and smoking did not show any significant association with the colonization rate. However, in the crude analysis, CTX prophylaxis, family size, $CD_4$ T cell counts, history of hospitalization, and gender were found to be associated. The odds of *S. pneumoniae* colonization in male HIV patients were more than two and a half times greater than that in females [COR: 2.7 (95% CI: 1.2–6)]. Prevalence of pneumococcal colonization is prominent (27.3%) among patients from rural area [COR: 2.7 (95% CI: 1.9–7.4)] and the prophylaxis increases the odds of colonization by more than one and a half fold in comparison to the reference category [COR: 1.7 (95% CI: 1.8–3.81)]. Lower $CD_4$ T cell counts (<250 cells/mm$^3$) were associated with a higher prevalence of pneumococcal colonization [COR: 2.35 (95% CI: 1.0–6.7)]. Although higher $CD_4$ T cell counts (350–500 cells/mm$^3$) were associated with a significant decrease in colonization among HIV patients as per bivariate analyses [COR: 0.6 (95% CI: 0.7–0.9)], it did not prove worthy in the case of multivariable models. Individuals living in a larger family seemed to be less likely to carry pneumococci than those who live in smaller ones [COR: 0.4 (95% CI 0.2–0.9)], which remains to be an odd finding and needs further in-depth analysis to check for other factors like the intimacy and frequency of contacts with children. Multiple logistic analyses revealed a predilection for male patients concerning the nasopharyngeal colonization. The odds of pneumococcal colonization among males was two and a half times higher than that found in females [AOR: 2.5 (95% CI: 1.09–5.93), p = 0.03].

**Table 2. Socio-demographic factors associated with pneumococcal colonization among HIV patients at Arba Minch Hospital, from April 01 to August 31, 2018 (n = 252).**

| Characteristics | | Total | Colonization No. (%) | COR (95%CI) | p-value | COR (95% CI) | P-value |
|---|---|---|---|---|---|---|---|
| **Age** | 16–29 | 35 | 6(17.1) | 0.33 (0.01–3) | 0.27 | - | |
| | 30–44 | 163 | 20(12.3) | 1* | | - | |
| | 45–59 | 52 | 7(13.5) | 0.15(0.01–2) | 0.27 | - | |
| | >60 | 1 | 1 | 0.14(0.01–2) | 0.29 | - | |
| **Gender** | Male | 144 | 26(18.0) | 2.7(1.2–6) | 0.14 | 2.5(1.09–5.93) | 0.03** |
| | Female | 108 | 8(7.4) | 1* | | 1* | |
| **Residence** | Urban | 230 | 28(12.2) | 1* | | 1* | |
| | Rural | 22 | 6(27.3) | 2.7(1.9–7.4)* | 0.04* | 1.09 (0.3–14) | 0.75 |
| **Smoking** | Yes | 17 | 3(17.6) | 1.2(0.4–5) | 0.60 | - | |
| | No | 235 | 31(13.7) | 1* | | - | |
| **Contact with children <5years** | Yes | 49 | 6 (12.2) | 0.87(0.3–2.2) | 0.77 | - | |
| | No | 203 | 28(13.8) | 1* | | - | |
| **Family size** | 1–3 | 118 | 22(18.8) | 1* | | 1* | |
| | >4 | 134 | 12(9.0) | 0.4(0.2–0.9) | 0.025* | 0.4(0.19-.89) | 0.25 |
| **Characteristics** | | **Total** | **Colonization No. (%)** | **COR (95% CI)** | **P-value** | **AOR (95%CI)** | **P-value** |
| **Hospitalization** | Yes | 14 | 3(12.5) | 1.0(0.4–6) | 0.37 | - | |
| | No | 238 | 31(13.5) | 1* | | - | |
| **ART** | Yes | 241 | 32(13.3) | 0.6(.14–3.3) | 0.63 | - | |
| | No | 11 | 2(18.2) | 1* | | - | |
| **History of URTI** | Yes | 45 | 4(8.88) | 1.12(0.4–3.03) | 0.86 | - | |
| | No | 207 | 30(14.49) | 1* | | - | |
| **History of LRTI** | Yes | 18 | 3(16.66) | 0.84(0.2–2.9) | 0.33 | - | |
| | No | 234 | 31(13.24) | 1* | | - | |
| **CD$_4$ T cell counts (cells/mm$^3$)** | >500 | 80 | 6(7.5) | 0.89(0.4–1.8) | 1* | 1* | |
| | 350–500 | 70 | 7(10.0) | 0.06(0.7–0.9) | 0.05* | 1.4(1.1–3.8) | 0.33 |
| | 250–299 | 62 | 9(14.51) | 1.35(1.0–1.7) | 0.76 | 2.1(1.14–3.8) | 0.78 |
| | < 250 | 40 | 12(30) | 2.35(1.0–6.7) | 0.25* | 2.2(1.4–3.6) | 0.08 |
| **WHO clinical stage** | Stage I | 85 | 6(7.05) | 1* | | - | |
| | Stage II | 41 | 7(17.07) | 1.13(1.2–2.4) | 0.38 | - | |
| | Stage III | 86 | 10(11.62) | 1.15(0.8–1.6) | 0.85 | - | |
| | Stage IV | 29 | 11(37.94) | 2.06(0.6–1.2) | 0.92 | - | |
| **Cotrimoxazole prophylaxis** | Yes | 126 | 21(16.7) | 1.7(1.8,3.6) | 0.140* | 2.2(1.05–4.9) | 0.037** |
| | No | 126 | 13(10.3) | 1* | | | |

CI: Confidence Interval, COR: Crude Odds Ratio; 1*: reference, ART: Antiretroviral Therapy, URTI: Upper Respiratory Tract Infection, LRTI: Lower Respiratory Tract Infection

* p-value less than 0.25, AOR: Adjusted Odds Ratio.

## Impact of cotrimoxazole prophylaxis on colonization rate of *S. pneumoniae*

Cotrimoxazole prophylaxis was an independent risk factor for pneumococci colonization and it doubled the risk of a sample being tested positive [AOR: 2.27 (95% CI: 1.05–4.9)] when the influence of other variables was adjusted for the confounder (Table 2).

## Antimicrobial susceptibility pattern

Among 34 isolates, a larger proportion was resistant; ie., 24(70.5%) to CTX, 17(50.0%) to penicillin, 24(61.7%); to tetracycline, 7(20.5%); to erythromycin, and 3(8.8%) to chloramphenicol.

**Table 3. Risk of colonization with drug-resistant pneumococci among study participants by their CTX prophylaxis status: A multivariate logistic regression analysis, Arba Minch, 2018 (n = 34).**

| Drug resistance pneumococci | Total | Percent of colonization | | COR (95% CI) | P-value | AOR (95% CI) | P-value |
|---|---|---|---|---|---|---|---|
| | | On CTX (n = 21) | Control (n = 13) | | | | |
| CTX | 24 | 18(85.7) | 6(46.2) | 7(1.3–36) | 0.01* | 6.7(1.3–36) | 0.02 |
| PEN | 17 | 11(52.4) | 6(46.2) | 1.3(0.3–5.1) | 0.22* | 1.3(1.2–4.3) | 0.05 |
| CAF | 3 | 2(9.5) | 1(7.7) | 1.2(0.1–15) | 0.67 | - | |
| ERY | 7 | 5(23.8) | 2 (15.4) | 1.7(0.2–10.5) | 0.68 | - | |
| TTC | 21 | 8(61.5) | 13(61.9) | 1.0(0.2–4.2) | 0.63 | - | |
| MDR | 10 | 8(38.1) | 2 (15.4) | 3.4(0.5–19) | 0.24* | 1.6(1.9–4.3) | 0.04 |

On CTX: Patient taking cotrimoxazole prophylaxis, PEN: Penicillin, CAF: Chloramphenicol, ERY: Erythromycin, TTC: Tetracycline, MDR: Multi-Drug Resistance, COR: Crude Odds Ratio.

All the isolates were susceptible to levofloxacin and vancomycin whereas susceptibility to other antibiotics varied considerably, with the highest being found against CTX and tetracycline. Multi-drug resistance was 29.5% only. The most common pattern of multi-drug resistance observed was towards penicillin, erythromycin, and CTX as shown in Table 3. Antimicrobial resistance to one or more antibiotics across the study groups was 90% and 74.6% respectively. The resistant isolates were found mainly in patients taking prophylaxis but were not statistically significant (Table 3).

## Effect of cotrimoxazole prophylaxis as a risk factor for antibiotic resistance

On analyzing the susceptibility patterns of pneumococci in the prophylaxis group concerning the control group, it has been found that 85.7 vs. 46.2% were CTX resistant; 52.4 vs. 46.2% were penicillin-resistant and 38.1 vs. 15.4% were MDR; in each case, the latter group is found to be less vulnerable. Multiple logistic regression analysis showed that HIV patients taking CTX prophylaxis were more likely to be colonized with pneumococci resistant to CTX (p-value = 0.03), penicillin (p-value = 0.05) and MDR (p-value = 0.04). A higher proportion, ie., 23.8% of isolates from patients taking prophylaxis showed resistance to erythromycin compared to 15.4% in the control group. The percentage of resistance to chloramphenicol was found to be minimal in the case of both groups, ie., 9.5 vs. 7.7%. Tetracycline resistant isolates were almost equal in percentage in patients taking CTX prophylaxis group and the control group, ie., 61.5 vs. 61.9% (Table 3).

**Table 4. Association of CTX resistance to other classes of drug co-resistance among pneumococci colonized study participants at Arba Minch, 2018: A multivariate logistic regression analysis.**

| Antibiotic | Total | No. of CTX resistance (%) | | COR (95%CI) | P-value | AOR (95%CI) | P-value |
|---|---|---|---|---|---|---|---|
| | | On CTX (n = 18) | Not on CTX (n = 6) | | | | |
| PEN | 13 | 11(61.1) | 2(33.3) | 5.4(1.1–2) | 0.21* | 4.6(1.6–4.9) | 0.02 |
| CAF | 2 | 2(11.1) | 0(0) | 1.2(0.4–1) | 0.85 | - | |
| ERY | 6 | 5(27.7) | 1(15.4) | 4.8(0.5–4) | 0.14* | 0.29(0.4–4) | 0.30 |
| TTC | 9 | 5(27.7) | 4(66.6) | 0.8(0.2–3) | 0.85 | - | |
| MDR | 10 | 8(44.4) | 2(33.3) | 0.6(0.4–0.8) | 0.81 | - | |

On CTX: Patient taking cotrimoxazole prophylaxis, CAF: Chloramphenicol, ERY: Erythromycin, PEN: Penicillin, TTC: Tetracycline, COR: Crude Odd Ratios, MDR: Resistance to one or more tested drugs.

### Effect of cotrimoxazole prophylaxis on co-resistance of pneumococci

The prevalence of cotrimoxazole and penicillin co-resistance was 11(61.1%) and 2(33.3%) in the CTX prophylaxis group and the control group respectively. The analysis showed that the former group is more likely to have cotrimoxazole and penicillin co–resistant isolates [AOR: 4.6 (95% CI: 1.6–49) p = 0.029]. Eight MDR isolates from the prophylaxis group were found to be co-resistant to CTX. Co-resistance rates to tetracycline and chloramphenicol were not statistically significant (Table 4).

**Risk of pneumococci colonization by follow-up time.**   Among the patients who had completed, an average 18 months of CTX prophylaxis (ranging from 1 to 36 months), poor adherence (missing three or more doses) was observed in the case of only three individuals (2.3%). Of the 126 patients belonging to the control group, six follow-up visits were recorded for 106 individuals (84.6%). One individual was excluded from the analysis because of the unknown date of commencement of prophylaxis.

There was an apparent increase in the number of CTX resistant and MDR pneumococci in patients under the prophylaxis scheme. Analysis of CTX resistant isolates by the duration of follow-up showed that CTX resistance increased from 28.5% during the first three months of prophylaxis to 85.7%, in a later stage, ie., in the 12th month and beyond. There was also evidence for a change in the rate of colonization of MDR *S. pneumoniae* over a period of cotrimoxazole prophylaxis i.e. from 10% colonization rate during the first three months of CTX prophylaxis to 33.3% in the 12th month and beyond. As per our analysis, during the 6th month of follow-up, 38% of pneumococci were found to be resistant to CTX in the prophylaxis group versus 31% in the control group. Multidrug resistance was estimated to be 24.7% in the CTX group against a nominal 2% in the control group (Fig 1). However, there was no apparent change in the colonization rate of *S. pneumoniae* over a time period in the case of the control group. Percentage changes in resistance to other antibiotics were considered only too marginal for any formal and meaningful analysis.

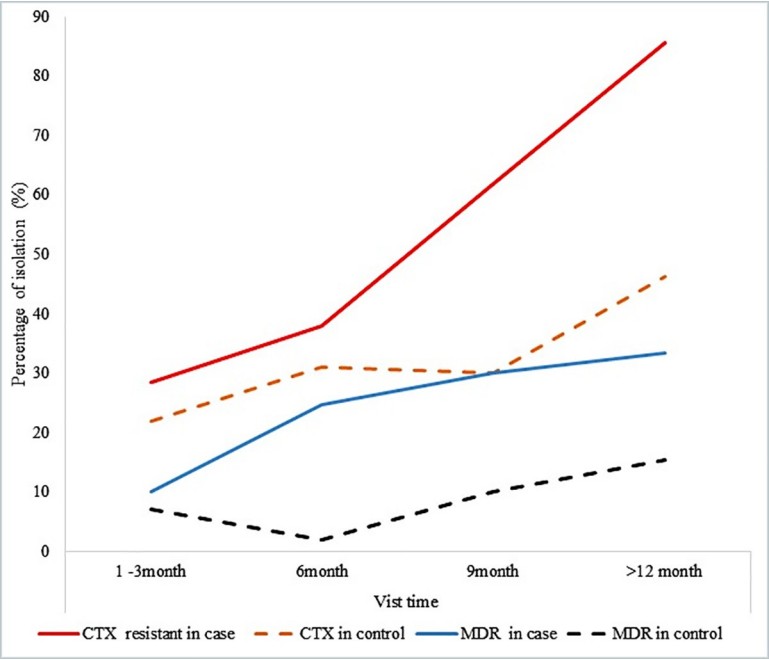

**Fig 1. Percent of CTX resistant and MDR pneumococci among colonized patients.**

## Discussion

This study has successfully demonstrated various aspects of the epidemiology of *S. pneumoniae* including the overall prevalence of nasopharyngeal colonization, associated factors, and antibiotic susceptibility patterns in HIV patients. It also aimed to assess the impact of cotrimoxazole prophylaxis on commensal bacteria and to alert policymakers /providers regarding the changes in drug resistance patterns.

Results revealed that the overall pneumococcal colonization is not that prominent ie., 13.47% only. Cotrimoxazole prophylaxis doubled the risk of carrying pneumococci [AOR: 2.27 per ≥ 12 month (95% CI: 1.05–4.9)]. CTX resistance and MDR colonization were pronounced in subjects taking CTX prophylaxis, in comparison to the control group (p-value = 0.03). Previous studies done in different regions of Africa have reported some consistent findings [6–8]. To the best of our knowledge, this is the first report describing the pneumococcal colonization and drug susceptibility patterns in HIV patients taking cotrimoxazole prophylaxis, as per the national policy of Ethiopia. The results are pertinent in the sense that the potential effects of widespread CTX prophylaxis must be well-monitored and further planning must be done soon.

### Nasopharyngeal colonization

The overall prevalence of pneumococcal colonization among HIV patients was 13.47% and it is matching to the previously reported prevalence of 10.8% in a similar group of adults found elsewhere [21], 18.0% in Uganda [37], 8.8% in South Africa [29] and 11.4% in Zambia [28]. The pneumococcal colonization rate (13.47%) that we had observed is also comparable to the prevalence rates, 20% and 22.% respectively reported from Malawi and Indonesia [8, 26]. However, it is higher than the 3% reported from the USA, 7.3% from Ghana, and 6.4% from Kenya [30–32]. At the same time, two studies reported a very high pneumococcal colonization rate ie., 43.2% and 34.6%, surprisingly both from Kenya [14, 38]. A possible explanation for these differences could be attributed to an alteration in the methodology applied, like the design of study, sampling, and also variations in the characteristics of sampled populations such as vaccination programs [3]. Colonization has significant implications since it provides a gateway to transmissions and invasive diseases.

HIV patients taking cotrimoxazole prophylaxis were two times more likely to be colonized by pneumococci than those who were not taking prophylaxis [AOR: 2.27 (95% CI: 1.05–4.9)]. This observation is in line with other previous studies undertaken in various parts of Africa [7, 8, 18, 19]. This appears to confirm the theory of selection of resistant mutants and clonal expansion related to the use of broad-spectrum drugs [13, 14]. However, some other works contrast the results of our study, the former showing no statistically significant association between pneumococcal colonization and cotrimoxazole prophylaxis [25]. Alternatively, the enhanced higher colonization rate among patients on CTX can be explained by low $CD_4$ T cell counts found in the respective group. Because, the mechanism of mucosal protection against pneumococcal colonization is T-cell dependent, HIV patients with low $CD_4$ T cell counts may be the group having the maximum risk of colonization [6]. However, no differences in the prevalence of colonization related to alteration in $CD_4$ T cell counts were previously found among HIV patients in other studies conducted in Kenya [37]. Also, it is to be mentioned that the present set of results are in contrast to a longitudinal study done in Zambia, which showed that CTX prophylaxis reduced the colonization by 7% [7].

### Antibiotic resistance

It was found that 94.1% of participants showed resistance to one or more antibiotics. This is higher compared to the 61% previously reported from a comprehensive study done in Gonder,

Ethiopia. Antibiotic resistance revealed a descending trend towards cotrimoxazole (70.5%) followed by tetracycline (61%), penicillin (50%), erythromycin (20.5%) and, chloramphenicol (8.5%). This finding is consistent with the general patterns of drug resistance observed in Zambia [29] and South Africa [28]; at the same time surprisingly some regions witnessed 0% resistance, ie., absolute susceptibility [7].

Cotrimoxazole resistant (85.7% vs. 46.2%) and MDR isolates (33.1 vs. 15.4) were more in number in samples from patients taking CTX prophylaxis than in the control group (p-value = 0.03). In accordance with the previous studies, our results confirmed that cotrimoxazole prophylaxis is an independent risk factor concerning the resistance shown by colonizing pneumococci [8, 16, 19, 28]. A higher percentage of CTX resistance found in this study is consistent with the results obtained from similar works done in Malawi (85%) [26], Kenya (92%) [6] and, Zambia (87%) [28]. Simultaneously there is an upward trend of percentage resistance to cotrimoxazole compared to the 52% reported from Uganda [20] and 60.9% from other parts of Africa [7, 19]. It could be depicted from the findings of this survey that CTX prophylaxis leads to pneumococcal resistance to the drug itself. This favors the hypothesis that pneumococci isolates have acquired resistance to cotrimoxazole due to selective pressure created by the widespread long term prophylaxis [19]. Increased colonization with cotrimoxazole resistant pneumococci substantiates the rising concerns that such prophylaxis may facilitate the emergence of drug-resistant organisms. However, our findings were in contrast to a previous study that reported only a negligible difference in the extent of CTX resistance shown by pneumococci triggered by the prophylaxis [16]. This mismatch may be due to the smaller sample size involved, fluctuations in follow-up time, and the level of dosage in the present study.

Besides, it has been observed that the proportion of cotrimoxazole resistance and penicillin co-resistance was 81.5% [AOR: 4.6 (95% CI: 1.6–4.9)]. This is in agreement with some previous findings from the USA and Africa [7, 33] and may also provide further evidence for the theory of evolution of pharmacological cross-tolerance between two different drugs across the globe. An alternative explanation for this is the co-selection of linked resistant genes [19, 21]. Penicillin-resistant pneumococci are of great concern, as severe infections in Africa are often treated with penicillin, even today [7, 19].

The colonization rate of multidrug drug-resistant pneumococci was significantly different among patients having CTX prophylaxis. This is consistent with a couple of previous reports [28, 33]. A plausible explanation for this might be that CTX resistant genes can also impart selective pressure for MDR. It is encouraging to observe that all pneumococci isolates were sensitive to levofloxacin and vancomycin (last line drug). This may be partly due to the fact that these antibiotics are there in the Ethiopian market for only a relatively shorter period of time due to national policies and regulations dealing with the use of antibiotics and hence not used extensively [39, 40].

A more generalized interpretation of the findings is that our data provide additional evidence that cotrimoxazole exposure has adverse microbiological consequences related to increased risk of pneumococci colonization resulting in the selection of resistance to cotrimoxazole, penicillin, and even multiple drugs. The observed colonization with cotrimoxazole resistant and MDR isolates in HIV patients taking prophylaxis may have implications for pneumococcal transmissions, causing the spread of resistance among family members and the community at large. In fact, the importance of the role of HIV patients in pneumococcal transmission was unclear. However, they may represent a large reservoir of *S. pneumoniae* which is not controlled by cotrimoxazole prophylaxis and ART. These findings may provide new insights into the effect of CTX prophylaxis on the evolution of drug-resistant commensals. Since the colonization of *S. pneumoniae* precedes invasive pneumococci diseases, it highlights the need for the pneumococci vaccine that is recommended by WHO. High CTX resistance,

CTX- penicillin dual resistance, and MDR by *S. pneumoniae* may be transferred to other common pathogens in the ecological niche. There exists a chance of using this baseline information for further larger-scale molecular study to probe the effect of CTX in driving away pneumococci from vaccine serotype. The successful roll-out of antiretroviral drugs has reduced the importance of CTX prophylaxis in developed countries. Therefore, we call for a thorough re-appraisal of the current policy to limit the extent of cotrimoxazole prophylaxis in HIV patients by augmenting the efforts to detect the infection promptly. This will allow for the timely introduction and wiser usage of antiretroviral treatment, thereby limiting the cotrimoxazole exposure in a large number of positive cases. Even though the lack of any resistance to levofloxacin and vancomycin is reassuring, an increase in the penicillin non-susceptibility is alarming.

This study is subject to some limitations. One of the principal limitations of this analysis is its confined cross-sectional nature. Employing only a standard manual culture method in isolating pneumococci may slightly underestimate the prevalence rate of nasopharyngeal colonization of pneumococci. Due to the shortage of reagents, isolates were not analyzed for capsular serotyping; even though this would not have influenced the comparison of pneumococcal colonization by CTX status, yet it might have affected the isolates that are similar in serotypes, which are targeted by the existing conjugate pneumococcal vaccines. Detection of antibiotic resistance and co-resistance to other classes of drugs by phenotypic methods can also influence the conclusion corresponding to the mechanism of resistance or cross-resistance. Hence our results could not be extrapolated to other representative bacterial populations in the respiratory tract. This warrants further molecular study to have a comprehensive genomic analysis of transferable resistance. We should also acknowledge that this study measures only the colonization at a single time-point, with a smaller sample size, that too in a particular institution only. This could have undermined the exact estimation of the prevalence rate. Moreover, the association of CTX exposure and colonization with drug-resistant pneumococci does not represent the characteristics of the general population since only HIV patients were included.

## Conclusions

Prevalence of nasopharyngeal pneumococcal colonization among HIV patients was 13.49% and this is in agreement with the findings of other studies undertaken in various parts of Africa. Colonization rate was associated with gender and CTX prophylaxis. The latter increased the risk of resistance to cotrimoxazole and even multiple drugs. Resistance to cotrimoxazole was statistically associated with co-resistance towards penicillin. These findings reinforce some of the previous concerns indicating that cotrimoxazole prophylaxis may worsen the already existing prevalence of drug resistance. It is therefore, imperative to limit the use of cotrimoxazole as a prophylactic drug only to those patients who will be benefited from it (WHO clinical stage II-IV or with $CD_4$ T counts less than 350 cell/mm$^3$). CTX prophylaxis is to be viewed with greater concern and must be correlated to extensive public health enlightenment and antibiotic stewardship programs controlling the spread of emerging drug-resistant superbugs. Continued monitoring of prevalent serotypes and antimicrobial resistance is important in estimating and evaluating the impact of CTX prophylaxis. Treatment of pneumococcal diseases also should be taken into account, in the light of the enhanced risk of resistance to antibiotics, especially among individuals taking cotrimoxazole prophylaxis.

## Supporting information

**S1 File. Subject information sheet.**
(DOCX)

**S2 File. Informed consent form.**
(DOCX)

**S3 File. Questionnaire.**
(DOCX)

**S1 Dataset.**
(XLS)

## Acknowledgments

We would like to thank Jimma University, College of Health Science, Department of Medical Laboratory Sciences, and Pathology. A very special thanks to the ethical review boards of Jimma University for giving ethical clearance. We would like to thank all the study participants, the data collectors, and the laboratory technicians who participated in the research work. Finally, we would like to record our special thanks to Dr. K.R. Sabu for rendering help in English corrections.

## Author Contributions

**Conceptualization:** Mohammed Seid.

**Data curation:** Mohammed Seid.

**Formal analysis:** Mohammed Seid.

**Investigation:** Mohammed Seid, Bereket Workalemahu, Mulugeta Delbo.

**Methodology:** Mohammed Seid, Getnet Beyene, Yared Alemu, Bereket Workalemahu, Mulugeta Delbo.

**Project administration:** Mohammed Seid, Getnet Beyene, Yared Alemu, Bereket Workalemahu, Mulugeta Delbo.

**Resources:** Mohammed Seid, Getnet Beyene, Yared Alemu, Bereket Workalemahu, Mulugeta Delbo.

**Software:** Mohammed Seid.

**Supervision:** Mohammed Seid, Getnet Beyene, Yared Alemu, Bereket Workalemahu, Mulugeta Delbo.

**Validation:** Mohammed Seid, Getnet Beyene, Yared Alemu, Bereket Workalemahu, Mulugeta Delbo.

**Visualization:** Mohammed Seid, Aseer Manilal.

**Writing – original draft:** Mohammed Seid, Getnet Beyene, Yared Alemu, Bereket Workalemahu, Mulugeta Delbo, Dagimawie Taddesse, Gelila Biresaw, Aseer Manilal.

**Writing – review & editing:** Mohammed Seid, Aseer Manilal.

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
