## [Decision Letter · Decision Letter 0]

30 Sep 2020

PONE-D-20-26041

Does cotrimoxazole prophylaxis in HIV patients increase the drug resistance of pneumococci ?:- A comparative cross-sectional study in southern Ethiopia

PLOS ONE

Dear Dr. Manilal,

Thank you for submitting your manuscript to PLOS ONE. After careful consideration, we feel that it has merit but does not fully meet PLOS ONE’s publication criteria as it currently stands. Therefore, we invite you to submit a revised version of the manuscript that addresses the points raised during the review process.

We look forward to receiving your revised manuscript.

Kind regards,

Grzegorz Woźniakowski, Full professor, PhD, ScD

Academic Editor

PLOS ONE

Journal Requirements:

2. Please include a separate caption for each figure in your manuscript.

Reviewers' comments:

Reviewer's Responses to Questions

**Comments to the Author**

1. Is the manuscript technically sound, and do the data support the conclusions?

Reviewer #1: Yes

2. Has the statistical analysis been performed appropriately and rigorously? 

Reviewer #1: Yes

3. Have the authors made all data underlying the findings in their manuscript fully available?

Reviewer #1: Yes

4. Is the manuscript presented in an intelligible fashion and written in standard English?

Reviewer #1: Yes

5. Review Comments to the Author

Reviewer #1: The article prepared by authors is very interesting and has a merit. The problem of HIV and multidrug resistance in bacteria is very important. The conclusions are supported by the results. All statistical analyses were conducted rigorously. Authors made tremendous work collecting all data. I have only a few small comments:

Title: remove “:-“

Line 62: use “,” instead of “;”

Line 81: write “AIDS”

Table 1: maybe remove religion part from the table, data could stigmatize some group of people

Line 285: “common” - the 6% difference is too low for saying that the pneumococcal colonization is common in compare to control group.

Line 449-451: lack of reference

Language in general is good however it could be enhanced.

I reccomend the article for publication after small (minor) revision.

Sincerely,

Reviewer

6. PLOS authors have the option to publish the peer review history of their article (what does this mean?). If published, this will include your full peer review and any attached files.

Reviewer #1: No

---

## [Author Response · Author response to Decision Letter 0]

14 Oct 2020

Responses to the Reviewers' comments

Reviewers’ comments were considered and carefully revised the article as per the suggestions and comments. We are very much thankful for the Reviewers’ for their comments and suggestions which had substantially improved the quality of the manuscript.

Editors comment Responses to the Reviewers' comments

Title: remove “:-“ Removed in the revised manuscript

Line 62: use “,” instead of “;” Included in the revised manuscript

Line 81: write “AIDS” Its not actually the word AIDS rather the verbal form of the word aid. And its corrected as adding in the revised manuscript

Table 1: maybe remove religion part from the table, data could stigmatize some group of people Removed in the revised manuscript

Line 285: “common” - the 6% difference is too low for saying that the pneumococcal colonization is common in compare to control group. Corrected as reasons” in the revised manuscript

Line 449-451: lack of reference Two new references have been added in the revised manuscript.

---

## [Decision Letter · Decision Letter 1]

16 Nov 2020

Does cotrimoxazole prophylaxis in HIV patients increase the drug resistance of pneumococci ?  A comparative cross-sectional study in southern Ethiopia

PONE-D-20-26041R1

Dear Dr. Manilal,

We’re pleased to inform you that your manuscript has been judged scientifically suitable for publication and will be formally accepted for publication once it meets all outstanding technical requirements.

Kind regards,

Grzegorz Woźniakowski, Full professor, PhD, ScD

Academic Editor

PLOS ONE

Additional Editor Comments (optional):

Reviewers' comments:

Reviewer's Responses to Questions

**Comments to the Author**

1. If the authors have adequately addressed your comments raised in a previous round of review and you feel that this manuscript is now acceptable for publication, you may indicate that here to bypass the “Comments to the Author” section, enter your conflict of interest statement in the “Confidential to Editor” section, and submit your "Accept" recommendation.

Reviewer #1: All comments have been addressed

2. Is the manuscript technically sound, and do the data support the conclusions?

Reviewer #1: Yes

3. Has the statistical analysis been performed appropriately and rigorously? 

Reviewer #1: Yes

4. Have the authors made all data underlying the findings in their manuscript fully available?

Reviewer #1: Yes

5. Is the manuscript presented in an intelligible fashion and written in standard English?

Reviewer #1: Yes

6. Review Comments to the Author

Reviewer #1: (No Response)

7. PLOS authors have the option to publish the peer review history of their article (what does this mean?). If published, this will include your full peer review and any attached files.

Reviewer #1: No

---

## [Editor Report · Acceptance letter]

26 Nov 2020

PONE-D-20-26041R1 

Does cotrimoxazole prophylaxis in HIV patients increase the drug resistance of pneumococci ? A comparative cross-sectional study in southern Ethiopia 

Dear Dr. Manilal:

I'm pleased to inform you that your manuscript has been deemed suitable for publication in PLOS ONE. Congratulations! Your manuscript is now with our production department. 

Kind regards, 

on behalf of

Prof. Grzegorz Woźniakowski 

Academic Editor

PLOS ONE